# Regularized inversion of aerosol hygroscopic growth factor probability density function: Application to humidity-controlled fast integrated mobility spectrometer measurements

Jiaoshi Zhang[1], Yang Wang[1,3], Steven Spielman[2], Susanne Hering[2], and Jian Wang[1]

[1]Center for Aerosol Science and Engineering, Washington University in St. Louis, St. Louis, Missouri, USA
[2]Aerosol Dynamics Inc, Berkeley, California, USA
[3]Department of Civil, Architectural and Environmental Engineering, Missouri University of Science and Technology, Rolla, Missouri, USA

*Correspondence to*: Jian Wang (jian@wustl.edu)

**Abstract.** Aerosol hygroscopic growth plays an important role in atmospheric particle chemistry and the effects of aerosol on radiation and hence climate. The hygroscopic growth is often characterized by a growth factor probability density function (GF-PDF), where the growth factor is defined as the ratio of the particle size at a specified relative humidity to its dry size. Parametric, least-square methods are the most widely used algorithms for inverting the GF-PDF from measurements of humidified tandem differential mobility analyzers (HTDMA) and have been recently applied to the GF-PDF inversion from measurements of the humidity-controlled fast integrated mobility spectrometer (HFIMS). However, these least square methods suffer from noise amplification due to the lack of regularization in solving the ill-posed problem, resulting in significant fluctuations in the retrieved GF-PDF and even occasional failures of convergence. In this study, we introduce nonparametric, regularized methods to invert aerosol GF-PDF and apply them to HFIMS measurements. Based on the HFIMS kernel function, the forward convolution is transformed into a matrix-based form, which facilitates the application of the nonparametric inversion methods with regularizations, including Tikhonov regularization and Twomey's iterative regularization. Inversions of the GF-PDF using the nonparameteric methods with regularization are demonstrated using HFIMS measurements simulated from representative GF-PDFs of ambient aerosols. The characteristics of reconstructed GF-PDFs resulting from different inversion methods, including previously developed least-square methods, are quantitively compared. The result shows that Twomey's method generally outperforms other inversion methods. The capabilities of Twomey's method in reconstructing the pre-defined GF-PDFs and recovering the mode parameters are validated.

## 1 Introduction

The hygroscopic growth of aerosol particles influences heterogeneous reactions, light extinction, and visibility, whereby aerosol water is most relevant for the direct radiative forcing of Earth's climate (Tang and Munkelwitz, 1994; Pilinis et al., 1995; Swietlicki et al., 2008). The ability of aerosols to absorb water depends mainly on their compositions, hence the hygroscopic properties reflect the variability of the key chemical components (Gysel et al., 2007; Zheng et al., 2020).

Therefore, the variation of aerosol hygroscopic growth can be used to infer the potential chemical composition, especially for small aerosols that are beyond the size range of the aerosol mass spectrometer. Aerosol hygroscopic growth under atmospheric relative humidity (RH) is commonly measured by a humidified tandem differential mobility analyzer system (HTDMA) (Liu et al., 1978; Rader and McMurry, 1986; Swietlicki et al., 2008). In an HTDMA system, monodisperse particles classified by the 1st DMA are exposed to an elevated RH in a humidity conditioner, and the size distribution of humidified particles is then measured by a 2nd DMA and a particle detector using scanning mobility technique. The particle hygroscopic growth is then derived from the size distribution of the humidified particles. Recently, a humidity-controlled fast integrated mobility spectrometer (HFIMS) was developed. The HFIMS replaces the 2nd DMA and particle detector within the HTDMA system with a water-based fast integrated mobility spectrometer (WFIMS), which captures the size distribution of humidified particles instantly (Pinterich et al., 2017a). As a result, the HFIMS drastically accelerates aerosol hygroscopic growth measurements (Pinterich et al., 2017b; Wang et al., 2019; Zhang et al., 2021), making it feasible to characterize ambient aerosol hygroscopic growth at a wide range of sizes and RH levels under ~ 25 min.

The HTDMA measurement, i.e., the mobility-concentration distribution of humidified particles, is a convolution of the aerosol hygroscopic growth factor probability density function (GF-PDF) and the transfer functions of both DMAs. Similarly, the HFIMS measurement represents a convolution of the aerosol GF-PDF together with the transfer functions of the DMA and the WFIMS (Wang et al., 2019). Two inversion algorithms, TDMAfit (Stolzenburg and McMurry (1988)) and TDMAinv (Gysel et al. (2009)) were developed and widely used to retrieve the GF-PDF from HTDMA measurements. In both algorithms, the GF-PDF is represented with a specific functional form, and the function parameters were derived by least-squares fitting. For example, the TDMAfit algorithm assumes the GF-PDF as a superposition of multiple Gaussian distribution functions (Stolzenburg and McMurry, 1988) or a summation of multiple lognormal (ML) distribution functions (Stolzenburg and McMurry, 2008). Likewise, TDMAinv describes the GF-PDF as a piecewise linear (PL) function at predefined growth factor values (Gysel et al., 2009). The function parameters are derived using least-squares fitting that minimizes the residual between the measured and reconstructed size distributions of humidified particles. Similar methods have been applied to invert GF-PDFs from HFIMS measurements by Wang et al. (2019).

Inversion of the GF-PDF from the HTDMA or HFIMS measurements is an ill-posed problem (Gysel et al., 2009). Least-squares methods such as TDMAfit and TDMAinv provide simple and effective ways to solve this ill-posed problem by representing the GF-PDF in a specific functional form (Kandlikar and Ramachandran, 1999). However, the GF-PDF inverted by the TDMAfit algorithm often relies on the initial guess of the parameters, resulting in occasional failures of convergence (Gysel et al., 2009). For example, it was reported that the TDMAfit algorithm may not be robust in cases of closely multiple overlapped modes and the successful convergence depends on the initial guess (Swietlicki et al., 2008). Moreover, it is well-known that the unregularized least-squares method amplifies the measurement noise (Kandlikar and Ramachandran, 1999; Sipkens et al., 2020), resulting in significant fluctuations in the retrieved GF-PDF. It has been shown that the derived GF-PDF using the TDMAinv algorithm may oscillate strongly when a higher bin resolution is chosen, while a too low resolution may not be adequate to reproduce complex shapes of true GF-PDF (Gysel et al., 2009). This may lead to incorrect interpretation of

the aerosol mixing state (Wang et al., 2019). The approach to overcoming noise amplification is to regularize the problem by including additional information, such as smoothness (Kandlikar and Ramachandran, 1999). Tikhonov regularization is among the most common regularization methods and has been applied to inversions of the aerosol size distribution (Talukdar and Swihart, 2003) and mass-mobility distribution (Sipkens et al., 2020). Recently, a software package was developed to invert HTDMA data using Tikhonov regularization (Petters, 2021). Twomey's method (Twomey, 1975), one of the most common

iterative regularization methods, has been widely used to invert aerosol size distributions (Collins et al., 2002; Olfert et al., 2008; Wang et al., 2018) and two-dimensional mass-mobility distributions (Rawat et al., 2016; Sipkens et al., 2020). However, to our best knowledge, Twomey's method has not been applied to invert GF-PDF from HTDMA or HFIMS measurements.

In this study, we present nonparametric, regularized inversions of the GF-PDF from HFIMS measurements. These inversion methods can be adapted to HTDMA measurements straightforwardly. The forward model (i.e., the convolution of the GF-

PDF, the transfer function of DMA, and the transfer function of WFIMS) is derived analytically and cast into a matrix form such that nonparametric inversion methods can be conveniently applied. The nonparametric inversions are demonstrated by retrieving GF-PDF from HFIMS measurements of ambient aerosols. The dependence of retrieved GF-PDF on GF bin resolutions is investigated, and an optimal GF bin resolution is identified. Synthetic data are generated using representative GF-PDFs of ambient aerosols and are applied to evaluate different inversion methods, including (1) parametric, least-squares

fittings, (2) nonparametric, unregularized least-squares, (3) Twomey's method, and (4) Tikhonov regularization. The performances of the different inversion methods including reconstruction accuracy, GF-PDF fidelity, smoothness, and computation time are presented and discussed.

## 2 Methods

This section presents the GF-PDF inversion routine from the HFIMS measurement, which includes the mathematical derivation

of the matrix-based inverse problem, the description of different inversion algorithms, and the generation of synthetic data for evaluating the inversion algorithms.

### 2.1 A matrix form for the forward model

The integrated response of HFIMS is determined by the aerosol size distribution, the DMA transfer function, the GF-PDF, and the WFIMS transfer function (Wang et al., 2019). The number concentration of particles with diameters between $D_{p1}$ and

$D_{p1} + dD_{p1}$ downstream of the DMA inside the HFIMS is given by:

$$dN_{\mathrm{DMA}} = \frac{Q_{\mathrm{a,DMA}}}{Q_{\mathrm{s,DMA}}} \eta_{\mathrm{chg}}(D_{\mathrm{p1}}) \eta_{\mathrm{p,DMA}}(D_{\mathrm{p1}}) \Omega(V_{\mathrm{DMA}}, \tilde{Z}_{\mathrm{p1}}) dN \tag{1}$$

where $Q_{\mathrm{a,DMA}}$ and $Q_{\mathrm{s,DMA}}$ are the DMA aerosol and sample (i.e., monodispersed) flow rates, respectively, $\eta_{\mathrm{chg}}(D_{\mathrm{p1}})$ is the aerosol charging efficiency, $\eta_{\mathrm{p,DMA}}(D_{\mathrm{p1}})$ is the particle penetration efficiency through the DMA, and $\Omega(V_{\mathrm{DMA}}, \tilde{Z}_{\mathrm{p1}})$ is the transfer function of the DMA operated with the classifying voltage of $V_{\mathrm{DMA}}$, $\tilde{Z}_{\mathrm{p1}}$ is the particle mobility ($Z_{\mathrm{p1}}$) normalized by

the DMA centroid mobility corresponding to $V_{\text{DMA}}$. $dN = n(D_{\text{p1}})dD_{\text{p1}}$ represents the number concentration of particles with diameters between $D_{\text{p1}}$ and $D_{\text{p1}} + dD_{\text{p1}}$. The number concentration of particles with diameters between $D_{\text{p2}}$ and $D_{\text{p2}} + dD_{\text{p2}}$ at the outlet of the conditioner is

$$dN_{\text{cond}} = dD_{\text{p2}} \int_{D_{\text{p1}}=0}^{D_{\text{p1}}=\infty} \eta_{\text{p,cond}}(D_{\text{p2}}) c_{\text{cond}}(D_{\text{p2}}, D_{\text{p1}}) dN_{\text{DMA}} \quad (2)$$

where the integration considers all possible values of $D_{\text{p1}}$. $\eta_{\text{p,cond}}(D_{\text{p2}})$ is the penetration efficiency of the conditioned particles, assuming the particle growth from $D_{\text{p1}}$ to $D_{\text{p2}}$ is instantaneous. $c_{\text{cond}}(D_{\text{p2}}, D_{\text{p1}})$ is the growth factor probability density function (GF-PDF) for particles with a dry diameter of $D_{\text{p1}}$ growing to a diameter of $D_{\text{p2}}$ during the humidity conditioning process. The GF-PDF satisfies $\int_{D_{\text{p2}}=0}^{D_{\text{p2}}=\infty} c_{\text{cond}}(D_{\text{p2}}, D_{\text{p1}}) dD_{\text{p2}} = 1$.

The WFIMS response to particles with diameters between $D_{\text{p2}}$ and $D_{\text{p2}} + dD_{\text{p2}}$ in the $i^{th}$ $D_{\text{p}}^*$ bin during any time interval ($t$) is calculated by

$$dR_i = \frac{Q_{\text{a,WFIMS}} N_F}{\dot{N}_F} \eta_{\text{p,WFIMS}}(D_{\text{p2}}) \eta_{\text{det}}(D_{\text{p2}}) \Omega_{\text{WFIMS},i}(Z_{\text{p2}}) dN_{\text{cond}} \quad (3)$$

$Q_{\text{a,WFIMS}}$ is the inlet flow rate through the WFIMS, $N_F$ is the number of frames being used to count $dR_i$, $\dot{N}_F$ is the frame rate. $N_F/\dot{N}_F$ represents the time interval ($t$) of counting, $\eta_{\text{p,WFIMS}}$ is the penetration efficiency of particles going through the WFIMS separator, $\Omega_{\text{WFIMS},i}(Z_{\text{p2}})$ is the transfer function of the $i^{th}$ bin of the instrument response diameter ($D_{\text{p}}^*$) of the WFIMS. Note that the detection efficiency for particles above 8 nm has been shown to be 1 (i.e., $\eta_{\text{det}}(D_{\text{p2}}) = 1$, Pinterich et al., 2017a).

The theoretical response of the $i^{th}$ $D_{\text{p}}^*$ bin of the HFIMS, $R_i$, can be derived by combining the above equations as detailed in Wang et al. (2019):

$$R_i = E \iint \frac{1}{D_{\text{p1}}} c_{\text{cond}}(g, D_{\text{p1}}) \Omega(V_{\text{DMA}}, \tilde{Z}_{\text{p1}}) \Omega_{\text{WFIMS},i}(Z_{\text{p2}}) dD_{\text{p2}} dD_{\text{p1}} + \epsilon_i \quad (4)$$

Where $E = R_{\text{tot}} \frac{b}{b_{\text{view}}} \frac{Q_{sh,\text{DMA}}}{Q_{a,\text{DMA}}} \frac{d\tilde{Z}_{\text{p1}}}{dD_{\text{p1}}}\Big|_{D_{\text{p1}}^*}$. $R_{\text{tot}}$ is the total counts of particles detected within the WFIMS viewing window, i.e., $R_{\text{tot}} = \sum_i R_i$, where $R_i$ is the response of the $i^{th}$ $D_{\text{p}}^*$ bin of the WFIMS. $b_{\text{view}}$ and $b$ are the length of the viewing area of the CCD-captured image and the length of the WFIMS mobility separator. $\epsilon_i$ is the error in the measured response. In Eq. (4), the GF-PDF is written as a function of growth factor $g$ (i.e., $D_{\text{p2}}/D_{\text{p1}}$), and it satisfies $c_{\text{cond,n}}(g, D_{\text{p1}}) dg = c_{\text{cond}}(D_{\text{p2}}, D_{\text{p1}}) dD_{\text{p2}}$. Given the narrow particle size range classified by the DMA, we assume the GF-PDF is the same for all particles classified by the DMA at a given voltage, i.e., $c_{\text{cond}}(g, D_{p1})$ is independent of $D_{\text{p1}}$ for the integration in Eq. (4). Rewriting the GF-PDF as $c_{\text{cond}}(g)$ and replacing $D_{\text{p2}}$ with $g D_{\text{p1}}$ in Eq. (4) gives:

$$R_i = E \int_0^{+\infty} dg\, c_{\text{cond}}(g) \int_0^{+\infty} dD_{\text{p1}} \Omega[V_{\text{DMA}}, \tilde{Z}_{\text{p1}}(D_{\text{p1}})] \Omega_{\text{WFIMS},i}[Z_{\text{p}}(g D_{\text{p1}})] + \epsilon_i \quad (5)$$

The integration can be approximated by a sum over $J$ GF bins, with the assumption that $c_{\text{cond}}(g)$ is a constant value within each GF bin:

$$R_{i,\text{theo}} = E \sum_{j=1}^{J} c_{\text{cond}}(g_j) \int_{g_{j-\frac{1}{2}}}^{g_{j+\frac{1}{2}}} dg \int_0^{+\infty} dD_{\text{p1}} \Omega[V_{\text{DMA}}, \tilde{Z}_{\text{p}}(D_{\text{p1}})] \Omega_{\text{WFIMS},i}[Z_{\text{p}}(gD_{\text{p1}})] + \epsilon_i \tag{6}$$

where $g_{j-1/2}$ and $g_{j+1/2}$ ($j = 1, 2, 3, ..., J$) are the lower and upper bounds of the $j$th GF bin. Eq. (6) can be further arranged into a matrix form (neglecting the error term) as

$$\mathbf{R} = \mathbf{M} \times \mathbf{c} \tag{7}$$

where the HFIMS response $\mathbf{R}$ is an $I \times 1$ array composed of $R_i$ ($i = 1, 2, 3, ..., I$). $I$ is the selected size bins of the WFIMS that covers the size range of $(0.8D_{\text{p1}}^*, 2.0D_{\text{p1}}^*)$ according to the settings of the DMA centroid diameter $D_{\text{p1}}^*$. The unknown GF-PDF $\mathbf{c}$, an $J \times 1$ array composed of $c_j$ ($j = 1, 2, 3, ..., J$), can be found by solving the Fredholm integral equation (7).

The element of the HFIMS kernel matrix, $\mathbf{M}$, is calculated by

$$M_{ij} = E_i \int_{g_{j-\frac{1}{2}}}^{g_{j+\frac{1}{2}}} dg \int_0^{+\infty} dD_{\text{p1}} \Omega[V_{\text{DMA}}, \tilde{Z}_{\text{p}}(D_{\text{p1}})] \Omega_{\text{WFIMS},i}[Z_{\text{p}}(gD_{\text{p1}})] \tag{8}$$

The HFIMS kernel describes the probability of particles with GF between $g_{j-1/2}$ and $g_{j+1/2}$ that is measured between the channel limits between $Z_{\text{p},i-1/2}^*$ and $Z_{\text{p},i-1/2}^*$. As described above, the inversion of the GF-PDF ($\mathbf{c}$) becomes an ill-posed problem due to overlapping of the HFIMS kernel function, like that of the aerosol size spectrometers (Kandlikar and Ramachandran, 1999; Collins et al., 2002; Talukdar and Swihart, 2003). It is worth noting that the derivation of the HFIMS kernel function can be easily applied to HTDMA measurement by replacing the WFIMS transfer function with the transfer function of the 2nd DMA in Eq. (8), as detailed in the supplementary information (SI).

## 2.2 Inversion methods

A number of techniques have been developed to solve the Fredholm integration (Kandlikar and Ramachandran, 1999). With Eqs. (7) and (8), nonparametric algorithms can be straightforwardly applied to invert GF-PDF, hence no prior knowledge of the functional form of GF-PDF is needed.

**Unregularized least-squares**

The simplest route is the ordinary least-squares (LSQ) which seeks to minimize the square of the residual:

$$\mathbf{c}^{\mathbf{LSQ}} = arg \min_{\mathbf{c}} \{\|\mathbf{Mc} - \mathbf{R}\|_2^2\} \tag{9}$$

where $\|\cdot\|_2$ denotes the Euclidean norm. Here, the least-squares solution is solved by using the *lsqnonneg* function from MATLAB. As the uncertainty in measurements can vary substantially for different $D_{\text{p}}^*$ bins, the residue is often weighted by measurement uncertainty. A weighted LSQ (WLSQ) seeks to minimize the weighted sum of squares (Sipkens et al., 2020):

$$\mathbf{c}^{\mathbf{WLSQ}} = arg \min_{\mathbf{c}} \{\|\mathbf{W}(\mathbf{Mc} - \mathbf{R})\|_2^2\} \tag{10}$$

where $\mathbf{W}$ denotes a diagonal weight matrix, whose $i^{\text{th}}$ diagonal element is the reciprocal of the standard deviation for data point $i$.

**Tikhonov regularization**

Tikhonov regularization is a common regularization method that overcomes noise amplification, and it has been used to invert aerosol size distribution and 2-D aerosol mass-mobility distributions (Talukdar and Swihart, 2003; Petters, 2021; Stolzenburg et al., 2022). In Tikhonov regularization, an additional regularization term is included in the least-squares approach:

$$\mathbf{c^{Tik}} = arg \min_{\mathbf{c}}\{\|\mathbf{Mc} - \mathbf{R}\|_2^2 + \lambda^2 \|\mathbf{Lc}\|_2^2\} \tag{11}$$

where $\lambda^2 \|\mathbf{Lc}\|_2^2$ represents the regularization term designed to minimize the derivative of a specific order and $\lambda$ is the regularization parameter that controls the degree of regularization. The penalization matrix $\mathbf{L}$ is often set as the identity matrix $\mathbf{I}$, the bidiagonal (-1, 1) matrix, and the upper tridiagonal (1, -2, 1) matrix for the $0^{th}$, $1^{st}$, and $2^{nd}$ order regularization, respectively (Hansen and O'Leary, 1993; Hansen, 1994). The parametric L-curve of $\|\mathbf{Mc}_\lambda - \mathbf{R}\|_2$ vs $\|\mathbf{Lc}_\lambda\|_2$ is plotted and the corner of the L-curve with the maximum curvature is identified using the "L-curve" routine from the regularization tools package developed by Hansen (1994). This optimal regularization parameter $\lambda$ corresponds to a good balance between minimization of the residual and reduction of the noise in the inverted $\mathbf{c}$ (Hansen, 1992; Hansen and O'Leary, 1993). Similarly, a weighted Tikhonov regularization (WTik) can be applied by (Sipkens et al., 2020):

$$\mathbf{c^{WTik}} = arg \min_{\mathbf{c}}\{\|\mathbf{W}(\mathbf{Mc} - \mathbf{R})\|_2^2 + \lambda^2 \|\mathbf{Lc}\|_2^2\} \tag{12}$$

The effect of introducing the weight in the LSQ inversion and Tikhonov regularization is examined in Section 3.2.

**Twomey's method**

Twomey's method is commonly used to find solutions for ill-posed problems and has been proved to be effective in inversions of the aerosol size distribution (Collins et al., 2002; Olfert et al., 2008) and aerosol mass-mobility distribution (Rawat et al., 2016; Sipkens et al., 2020). It is a non-linear optimization method and provides iterative regularizations. An initial guess solution is iteratively multiplied by small multiples of the HFIMS kernel function which are proportional to the ratio of the measured to calculated measurements as follows:

$$c_j^{k+1} = \left[1 + \left(\frac{R_i}{\mathbf{m}_i\mathbf{c}^k} - 1\right) M_{ij}\right] \cdot c_j^k \tag{13}$$

where $\mathbf{m}_i$ is the $i$th row of the HFIMS kernel function $\mathbf{M}$, and $R_i/\mathbf{m}_i\mathbf{c}^k$ denotes the relative divergence between actual and reconstructed HFIMS measurements. The positively constrained, least-squares solution is set as the initial guess (Olfert et al., 2008). Then, the initial guess is smoothed using a three-term moving average (Markowski, 1987) and input into the iterative Twomey's routine which is then repeated until a Chi-squared criterion is satisfied. It is worth noting that Twomey's method may require sufficient counting statistics to ensure converged solutions.

**Parametric LSQ fittings**

The parametric fitting methods assume a prior known distribution of the GF-PDF and calculate the forward model problem (Eq. 4) to reconstruct the HFIMS measurements. A nonlinear least-squares fitting with boundary constraints is performed to

search for the least-squares solution within the bounds. The ML and PL fitting routines for the GF-PDF inversion from HFIMS measurements have been developed by Wang et al. (2019). The influence of counting statistics and GF-PDF parameters (i.e., the number of modes of ML GF-PDF and the number of sections of PL GF-PDF) has been statistically studied. In this work, the GF-PDF inverted using ML and PL fitting routines with the optimized parameters are compared with those retrieved using nonparametric inversion methods described above.

## 2.3 Generation of synthetic data to evaluate inversion algorithms

HFIMS measurements are synthesized to evaluate the performance of different inversion methods. The synthetic data are based on three representative GF-PDFs that consist of one, two, and three lognormal modes, respectively. The mode parameters of the pre-defined GF-PDFs are listed in Table 1, similar to those listed in Wang et al. (2019). The parameters of $f$, $G$, $\sigma$ are the fractional weight, mean diameter growth factor, and geometric standard deviation of each mode. The theoretical HFIMS response (i.e., $R_i$) is derived using Eq. (4) based on each of the three GF-PDFs, and Gaussian and Poisson noise are then added to the response using the following approach. First, a zero-mean Gaussian noise component is added to the theoretical HFIMS response to simulate the system noise such as fluctuation of the sample flow rate:

$$R_{i,\mathrm{G}} = R_i(1 + \alpha n_i^\mathrm{G}) \tag{14}$$

where $R_i$ is the derived theoretical response of the $i^\mathrm{th}$ $D_\mathrm{p}^*$ bin, $n_i^\mathrm{G}$ is the $i^\mathrm{th}$ element of a standard normally-distributed random vector, $n^G$, with zero mean and variance of 1. The magnitude of the Gaussian noise is varied using a factor, $\alpha$. The HFIMS measurement is then simulated using the following Poisson distribution to reflect the discrete nature of the particle counting process:

$$P(x) = \frac{R_{i,\mathrm{G}}^x}{x!} \exp\left(-R_{i,\mathrm{G}}\right) \tag{15}$$

where $P(x)$ is the probability that $x$ number of particles are detected by HFIMS in the $i^\mathrm{th}$ $D_\mathrm{p}^*$ bin (i.e., actual measurements). The impact of the Gaussian noise on the performance of the inversion methods is examined for different noise levels in Section 3.2. Five hundred sets of HFIMS measurements are generated using Monte Carlo methods with constant counting statistics (i.e., $R_\mathrm{tot}$ of 100). These synthetic HFIMS measurements are then used to evaluate the inversion methods described above. Note that in the forward model for deriving the theoretical HFIMS response (i.e., Eq. 4), a higher resolution of $g$ (i.e., 120 bins over 0.8 - 2.0) is used than that of the HFIMS kernel matrix (i.e., 20 bins of $g$, Eq. 8). The difference between the forward and inverse models, together with the inclusion of Gaussian and Poisson noises, minimizes the effect of inverse crime (Colton et al., 1998).

**Table 1.** Mode parameters of representative GF-PDFs for generating synthetic HFIMS measurements.

| Predefined GF-PDF | Mode 1 | | | Mode 2 | | | Mode 3 | | |
|---|---|---|---|---|---|---|---|---|---|
| | $f$ | $G$ | $\sigma$ | $f$ | $G$ | $\sigma$ | $f$ | $G$ | $\sigma$ |
| 1 | 1.0 | 1.40 | 1.15 | | NA | | | NA | |

| 2 | 0.45 | 1.10 | 1.05 | 0.55 | 1.30 | 1.05 | | NA | |
| 3 | 0.39 | 1.05 | 1.10 | 0.32 | 1.40 | 1.05 | 0.29 | 1.70 | 1.10 |

## 3 Results and discussion

### 3.1 Optimal numbers of Growth factor bins and HFIMS size bins ($D_p^*$)

The numbers of GF bins ($J$) and $D_p^*$ bins ($I$) determine the dimensions of HFIMS kernel function, which affects the inversion of GF-PDF. The optimal number of $D_p^*$ bin is a trade-off between sizing resolution and counting statistics. Wang et al. (2019)

examined the influence of WFIMS $D_p^*$ bin number ($I$) on the inverted GF-PDF and found an optimal range of 23-32 for total particle counts of 100. For representative remote continental and urban aerosols, the number of particles measured by the HFIMS often exceed 100 in 20 seconds (Pinterich et al., 2017b; Zhang et al., 2021), ensuring sufficient counting statistics for ambient measurements. The dynamic range of WFIMS is roughly a factor of 10 in mobility, corresponding to a factor of ~3 in the size range (Zhang et al., 2021). In this study, 30 size bins (i.e., $I = 30$) that are evenly spaced on a logarithmic scale over

the WFIMS size range are used in the inversions.

The influence of growth factor bin number ($J$) on the inverted GF-PDF is examined using the synthetic HFIMS measurements described above. The GF-PDF was inverted from each set of the simulated HFIMS measurements using different GF bin numbers ranging from 10 to 50 (i.e., corresponding to a GF resolution range of 0.024 – 0.12). To facilitate the comparison of GF-PDFs inverted with different GF bin numbers, we interpolate the inverted GF-PDFs to 120 fixed growth factors that are

225 evenly distributed from 0.8 to 2.0. The average error of the inverted GF-PDF $\gamma$ is defined as:

$$\gamma^2 = \frac{1}{N}\sum_{i=1}^{N}\left(c_{i,inv} - c_{i,sim}\right)^2 \tag{16}$$

where $c_{i,inv}$ and $c_{i,sim}$ are the interpolated GF-PDF and pre-defined GF-PDF (i.e., true values) at the 120 fixed growth factors, respectively. $N$ is the number of points of fixed growth factors (i.e., 120). The smoothness of the inverted GF-PDF is evaluated using the absolute second-order derivative:

$$\xi = \sum_{i=2}^{N-1}|2c_{inv}(g_i) - c_{inv}(g_{i+1}) - c_{inv}(g_{i-1})| \tag{17}$$

To evaluate how well the inverted GF-PDF reproduces the HFIMS measurement, we define the residual of the reconstructed HFIMS measurement (i.e., reconstruction error) as:

$$\chi^2 = \sum_{i=1}^{L}\left(\tilde{R}_{i,inv} - \tilde{R}_i\right)^2 \tag{18}$$

where $\tilde{R}_{i,inv}$ is the normalized HFIMS measurement that is reconstructed using Eq. (7) (i.e., forward calculation). $\tilde{R}_i$ is the

235 normalized synthetic HFIMS measurement (i.e., true values).

Figure 1 shows the smoothness of the inverted GF-PDF ($\xi$) versus the residual of reconstructed HFIMS measurement ($\chi^2$) for different GF bin numbers ($J$). The variation of $\xi$ with $\chi^2$ exhibits an L-shaped curve for all three representative PF-PDF. The initial increase of $J$ from 10 to 20 substantially improves the agreement between the reconstructed and simulated HFIMS

measurements, as indicated by a much reduced $\chi^2$ value. At the same time, $\xi$ remains relatively small, indicating a high

smoothness of the inverted GF-PDF. In contrast, an increase of $J$ above 20 leads to a minor reduction of $\chi^2$ value but a drastic

increase of $\xi$, suggesting strong noise in the inverted GF-PDF. The optimal solution lies near the corner of the "L-curve"

(Hansen and O'Leary, 1993) that strikes a balance between the smoothness and the fidelity to the HFIMS measurements. For

all three pre-defined GF-PDFs, the corner of the L-curve corresponds to a $J$ value of 20. GF-PDF inverted with 20 growth

factor bins generally shows the smallest error ($\gamma^2$), indicating best agreements between the inverted and the true GF-PDFs.

Note that the above results are based on inversions using Twomey's method. The same type of L-curves for GF-PDFs inverted

using unregularized LSQ and Tikhonov regularizations are shown in SI (Section S2), and they also reveal a corner that

corresponds to a $J$ value of 20. These results suggest an optimal $J$ value of 20 for a range of representative GF-PDFs and

different inversion methods.

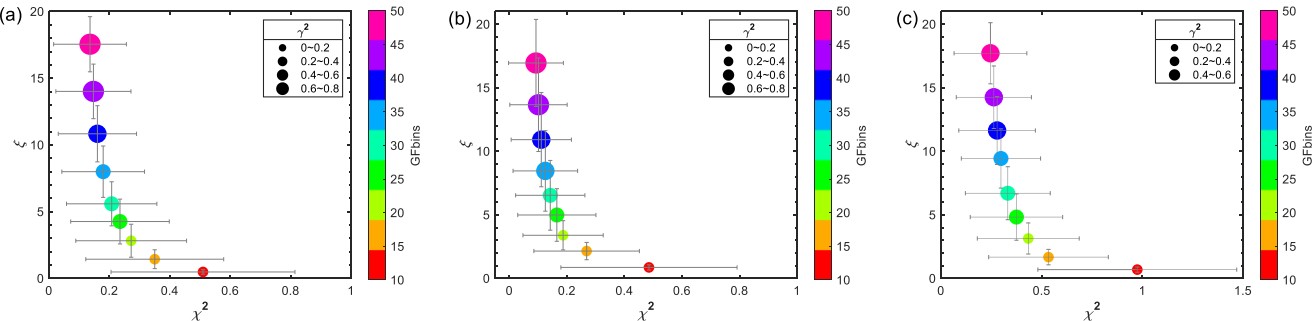

**Figure 1.** "L-curve" showing the dependence of reconstruction residual, $\chi^2$, and the smoothness, $\xi$, on the number of GF bins of pre-defined
GF-PDFs with (**a**) one mode, (**b**) two modes, and (**c**) three modes, respectively. The symbol size represents the error in inverted GF-PDF,
$\gamma^2$. Whiskers represent standard deviation. The inversion is conducted using Twomey's method.

### 3.2 Effect of measurement uncertainties

The uncertainty in HFIMS measurements consists of mainly normal distributed random instrumental noise (e.g., sample flow

fluctuation) and Poisson noise due to counting statistics. As the uncertainty varies among different HFIMS $D_p^*$ bins, we first

compare the performance of weighted and unweighted inversion methods, including LSQ and Tikhonov regularizations. For

this comparison, inversion methods are applied to HFIMS data synthesized with $\alpha$=0.05, a typical value used in previous

studies (Gysel et al., 2009). A total of 500 sets of synthetic data are generated for each of the three pre-defined GF-PDFs. The

values of synthesized HFIMS response ($R_{i,s}$) are integers, which reflect the discrete nature of particle counting. For weighted

LSQ and Tikhonov regularizations, the weight for $D_p^*$ bins (i.e., diagonal elements in **W**) is derived as $1/\sqrt{R_{i,s}}$. However, this

approach leads to a weight of infinite when $R_{i,s}$ has a value of zero (i.e., no particle detected within the $D_p^*$ bin). To overcome

this issue, we replace zero $R_{i,s}$ values with a fixed number $R_{i,\min}$ when deriving the weight. Figure 2 compares the

reconstruction residual, the GF-PDF error, and the smoothness of GF-PDF inverted using unweighted LSQ and weighted LSQ

with $R_{i,\min}$ values of 1, 0.1, 0.01, respectively. Whereas statistically no substantial difference is found among the smoothness

of GF-PDFs inverted using unweighted and weighted LSQ, unweighted LSQ leads to lower reconstruction residual and the error in inverted GF-PDF compared to the weighted LSQ. For the weighted LSQ inversions, both the reconstruction residue and the error in inverted GF-PDF increase with increasing weight for $R_{i,s}$ of zeros values (i.e., $1/\sqrt{R_{i,\min}}$). The measurement uncertainty is larger and therefore the weight is lower for channels with higher $R_{i,s}$, which corresponds to higher probability densities (i.e., higher $c(g)$ values). As a result, the GF-PDF inverted using weighted LSQ may have relatively larger errors for

high $c(g)$ values, and consequently the average GF-PDF error ($\gamma^2$). The same comparisons are also carried out for weighted and unweighted Tikhonov algorithms, and again the weighted algorithms do not provide better performances (i.e., lower error in inverted GF-PDFs) than the unweighted ones. Therefore, subsequent analyses of this study are focused on unweighted algorithms for LSQ and Tikhonov regularizations. It is worth noting that derivation of the weight as $1/\sqrt{R_{i,s}}$ implicitly assumes that the noise in HFIMS measurements is due to counting statistics only, whereas the synthetic HFIMS data are generated with

5% Gaussian noise. As shown next, the noise in the synthetic HFISM data is dominated by the counting statistics. In addition, for real measurements, the level of Gaussian noise (i.e., $\alpha$) is often not accurately known. We also repeated the above comparisons by deriving the weight as $1/\sqrt{(R_{i,s} + \alpha^2 R_{i,s}^2)}$, which accounts for both Poisson and Gaussian noises. The results are essentially the same.

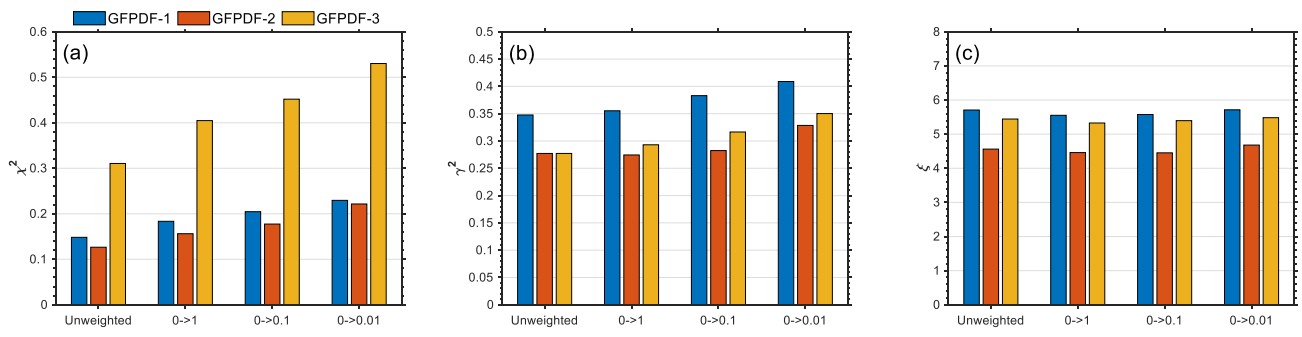

**Figure 2.** Comparison of reconstruction residual, $\chi^2$ (**a**), the GF-PDF error, $\gamma^2$ (**b**), and the smoothness, $\xi$ (**c**) of GF-PDFs inverted using the unweighted and weighted LSQ methods with different weighting schemes for zero value $D_p^*$ bins (i.e., replacing zero values by 1, 0.1, and 0.01, respectively). Colors correspond to the pre-defined GF-PDFs with one mode (blue), two modes (orange), and three modes (yellow). The results are averages based on inversions of 500 sets of synthetic HFIMS data for each of three pre-defined GF-PDFs.

The effect of the level of Gaussian noise on the inverted GF-PDF is examined. Synthetic HFIMS measurements are generated following the approach described above (Eq. 13 and 14) at four Gaussian noise levels (i.e., $\alpha$= 0%, 1%, 5%, and 10%). At each $\alpha$ level, 500 sets of synthetic data are generated and inverted using Twomey's method for each of the three pre-defined GF-PDF. All retrieved inversion parameters, including the reconstruction residual, the GF-PDF error, and the smoothness, are statistically the same for all four Gaussian noise levels (Fig. 3), indicating that HFIMS measurements noise is dominated by

counting statistics, and the inclusion of the Gaussian noise has negligible impact on the GF-PDF inverted by Twomey's

method. Similarly, the impact of Gaussian noise is also negligible for GF-PDF inverted using unweighted LSQ and $0^{th}$, $1^{st}$, and $2^{nd}$ order Tikhonov regularizations (not shown).

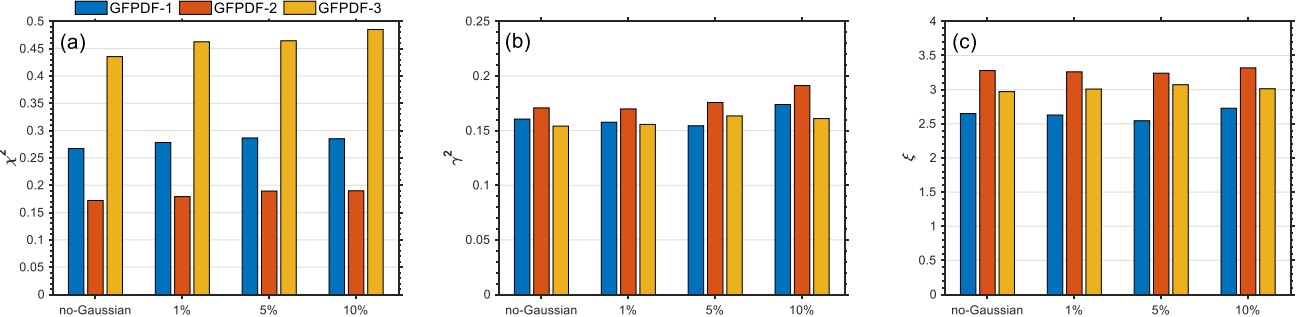

**Figure 3.** Comparison of reconstruction residual, $\chi^2$ (**a**), the GF-PDF error, $\gamma^2$ (**b**), and the degree of smoothing, $\xi$ (**c**) of GF-PDFs inverted using Twomey's methods from synthetic HFIMS data with additional Gaussian noises of different levels (i.e., none, 1%, 5%, and 10%). Colors correspond to the pre-defined GF-PDFs with one mode (blue), two modes (orange), and three modes (yellow). The results are averages based on inversions of 500 sets of synthetic HFIMS data for each of three pre-defined GF-PDFs.

We also challenged the inversion algorithms with different forward and inverse models to simulate the scenarios when DMA or WFIMS is not perfectly calibrated. A different DMA or WFIMS transfer function width is used to generate the synthetic HFIMS measurements than that used to calculate the inversion matrix. We found that up to ±20% variation of the DMA or WFIMS transfer function width has negligible impacts on the inverted GF-PDF. The results and discussion are detailed in Section S3 of the SI.

### 3.3 Comparisons of different inversion methods

The performances of different inversion methods described in Section 2.2 are systematically compared. A total of 500 sets of synthetic HFIMS data are generated and inverted for each of three pre-defined GF-PDFs. For all nonparametric methods, the inversions were carried out using the optimal numbers of GF bins ($J$) and $D_p^*$ bins ($I$), 20 and 30, respectively. Figure 4 shows the residual of reconstructed HFIMS measurements ($\chi^2$), the smoothness ($\xi$), the error of inverted GF-PDF ($\gamma^2$), and the computing time for different inversion methods. Compared with parametric counterparts (i.e., ML and PL least-squares fitting), the nonparametric methods generally retrieve more accurate GF-PDFs. Note that the ML least-squares fitting fails to converge to a valid solution occasionally, resulting in the abnormally large error in the inverted GF-PDFs, particularly for the pre-defined GF-PDFs with two and three modes. It may be due to the assumed spectral shape of GF-PDFs or the finite range of the boundary constraints that lead to a failure of searching for a least-squares solution in the presence of random noise. Among all nonparametric inversion methods, the unregularized LSQ provides the solution with the lowest reconstruction residual but largest noise and error in the inverted GF-PDFs, consistent with the noise amplification in unregularized methods. In comparison, regularized inversion methods generally produce smoother solutions at the expense of increased reconstruction

residuals. Among different Tikhonov regularization methods, higher-order regularizations (i.e., $1^{st}$ and $2^{nd}$) tend to produce smoother solutions, although the errors in inverted GF-PDF are very similar statistically. The $\xi$ value of the GF-PDF inverted using $1^{st}$ and $2^{nd}$ order Tikhonov regularizations increase with the mode number of GF-PDF, consistent with the increasingly

more complex spectral shape of GF-PDF. Overall, Twomey's method outperforms the Tikhonov regularization methods regardless of the shapes of the pre-defined GF-PDFs. On average, the GF-PDF inverted using Twomey's method has the smallest error ($\gamma^2$) and lowest $\xi$ value, indicative of the best performance. Note that the results are based on synthetic data generated with relatively low counting statistics (i.e., $R_{tot}$ of 100). We also synthesized HFIMS data with $R_{tot}$ of 500 and compared the performance of different inversion methods for measurements with the improved counting statistics, and the

results are consistent with those shown in Fig. 4 (Fig. S7). We, therefore, expect the results reflect the general performances of different inversion methods for a typical range of counting statistics of HFIMS measurements.

Figure 4(d) shows that once the matrix is generated, the implementation of the nonparametric methods requires a much shorter computing time than the parametric fitting methods. Here, the computing time is recorded on a desktop with Intel's 8th generation processor Core i7-8700. On average, a single-time implementation of the unregularized LSQ (i.e., the "lsqnonneg"

function in MATLAB) requires ~1s for all three pre-defined GF-PDFs, and the computing times for all other nonparametric methods are similar (with the largest difference of only ~4%), indicative of equally good computing efficiencies. In contrast, both ML and PL least-squares fitting routines require more than one order of magnitude longer time.

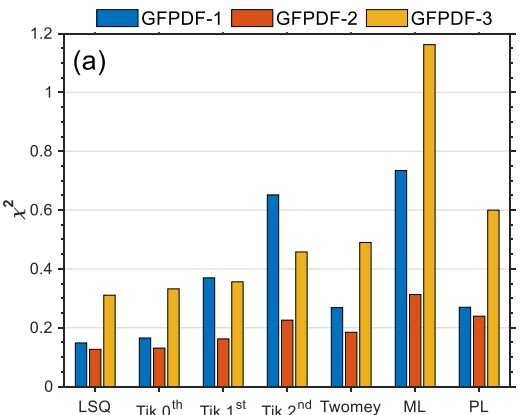
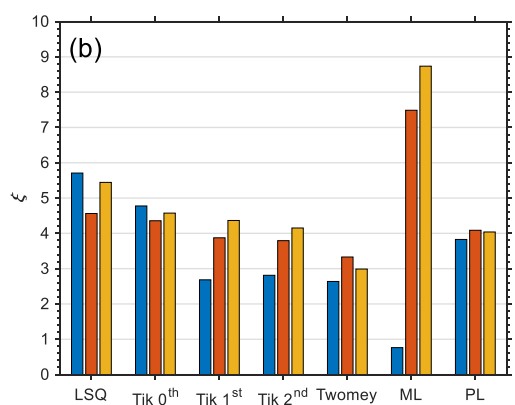

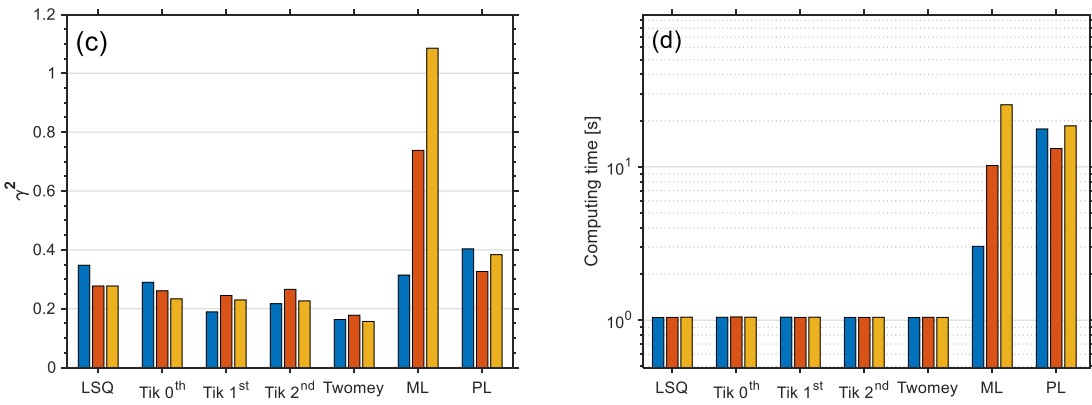

**Figure 4.** Comparison of reconstruction residual, $\chi^2$ (**a**), the smoothness, $\xi$ (**b**), the GF-PDF error, $\gamma^2$ (**c**), and the computing time (**d**) of GF-PDFs inverted using different inversion methods. Colors correspond to the pre-defined GF-PDFs with one mode (blue), two modes (orange), and three modes (yellow). The results are averages based on inversions of 500 sets of synthetic HFIMS data for each of three pre-defined GF-PDFs.

### 3.4 Comparison of Tikhonov regularization and Twomey's method

In this section, we investigate why Twomey's method performs better than Tikhonov regularizations. The Tikhonov regularized solutions depend on the regularization parameter, $\lambda$. The value of $\lambda$ is often determined by heuristic methods, including the L-curve approach (Hansen and O'Leary, 1993) and the Hanke-Raus rule (Hanke and Raus, 1996). The L-curve approach determines $\lambda$ by seeking a trade-off between minimizing the residual term and minimizing the regularization term (i.e., roughness of the solution), and the Hanke-Raus rule selects a computable $\lambda$ that minimizes the $\lambda$-dependent residual term $\frac{1}{\lambda}\left\|\mathbf{M}\mathbf{c}^{Tik}(\lambda) - \mathbf{R}\right\|_2$ (Hanke and Raus, 1996; Sipkens et al., 2020). As the pre-defined GF-PDFs are known for the synthetic HFIMS data, the value of $\lambda$ can be optimized by comparing the inverted GF-PDF with the true solution, i.e., minimizing the error in inverted GF-PDF ($\gamma^2$). Figure 5 shows the comparison of the statistics of inversions using LSQ, Twomey's method, and 1st order Tikhonov. The results are averages based on inversions of 500 sets of synthetic HFIMS data for each of three pre-defined GF-PDFs. Here, the 1st order Tikhonov regularization is chosen as it shows better performance (i.e., lower GF-PDF error) than 0th and 2nd Tikhonov regularizations (Fig. 4). The Tikhonov regularization parameter is identified by all three methods: (1) the L-curve, (2) the Hanke-Raus rule, and (3) optimization through minimizing the error in inverted GF-PDFs. It is worth noting that the 3rd method is not feasible for real measurements, as the true GF-PDF is unknown. Figure 5b shows that the Tikhonov regularization with the optimized $\lambda$ (i.e., the 3rd method) provides the most accurate solution (i.e., lowest GF-PDF error), and outperforms Twomey's method. However, when $\lambda$ derived using the L-curve approach or Hanke-Raus rule is used, GF-PDF inverted using 1st order Tikhonov regularization generally has a larger error (i.e., $\gamma^2$) than that inverted using Twomey's method. The above comparisons indicate that while the Tikhonov regularization can outperform Twomey's method in theory, the optimal regularization parameter $\lambda$ cannot be obtained reliably using existing methods in practice,

leading to inferior performance than Twomey's method. For example, the L-curve approach does not work well if the curvature of the L-curve is negative everywhere, and in such scenario, the leftmost point (i.e., with smaller $\lambda$) on the L-curve is taken as the corner (Hansen, 1994), leading to insufficient regularizations of the solution (Naseri et al., 2021). On the other hand, the Hanke-Raus rule often chooses a much larger $\lambda$ compared with the optimal value, which results in over-smoothed solutions with even larger errors. We also carried out similar comparisons of Twomey's method with $0^{th}$ and $2^{nd}$ order Tikhonov regularizations using $\lambda$ values derived from the three different methods, and the results are consistent with those shown in Fig. 5.

The nonparametric inversion methods are also applied to HFIMS measurements of ambient particles with a dry diameter of 35 nm (Zhang et al., 2021), as detailed in the SI (Section S5). As the true GF-PDF of ambient aerosols is unavailable, the performance of the inversion methods can not be directly compared. Nevertheless, the comparison of the reconstruction residual and the smoothness of inverted GF-PDF paints a similar picture that Twomey's method strikes a good balance between the smoothness of the inverted GF-PDF and the fidelity in reproducing the HFIMS measurements, and it likely outperforms Tikhonov regularizations in practice.

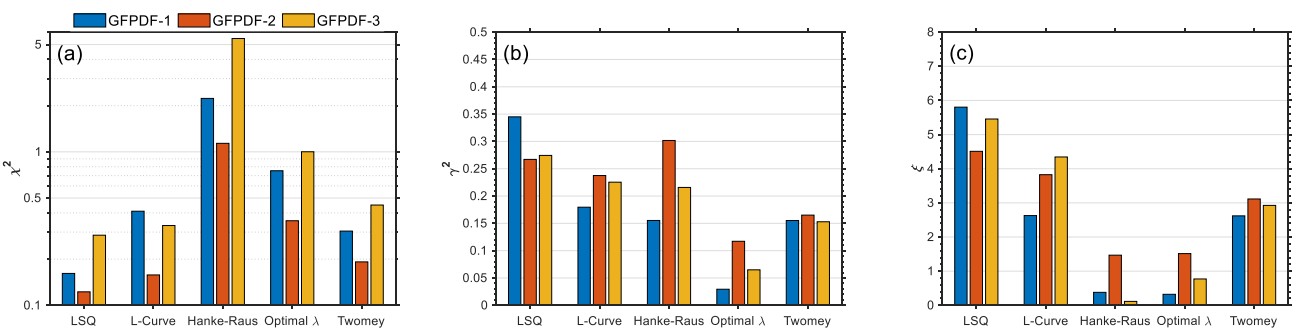

**Figure 5.** The reconstruction residual, $\chi^2$ (a), the GF-PDF error, $\gamma^2$ (b), and the smoothness, $\xi$ (c) of GF-PDF inverted using LSQ, $1^{st}$ order Tikhonov regularization with the regularization parameter derived from three different approaches (L-curve, Hanke-Raus rule, and optimized $\lambda$), and Twomey's method. The colors correspond to the pre-defined GF-PDFs with one mode (blue), two modes (orange), and three modes (yellow).

### 3.5 Inversion by Twomey's method

As Twomey's method is shown to be the best among all inversion methods examined, we characterize the accuracy of the GF-PDFs inverted using Twomey's method and the recovered mode parameters. Figure 6 compares the GF-PDFs inverted with the optimized GF and $D_p^*$ bin numbers and with the pre-defined GF-PDFs. The reconstructed and the simulated HFIMS measurements are also presented in the top panel. Both the inverted GF-PDF and reconstructed HFIMS measurements are averaged over the inversions of 500 sets of synthetic data. The results demonstrate excellent agreement of the reconstructed HFIMS measurements with the synthetic data (i.e., simulated HFIMS measurements) for all three pre-defined GF-PDFs. Both the spectral shapes and peak locations of the inverted GF-PDFs agreed well with that of the pre-defined GF-PDFs. In addition,

the inverted GF-PDFs are also in better agreement as compared with those inverted from parametric least-squares approaches

(i.e., ML and PL GF-PDFs, Wang et al., 2019).

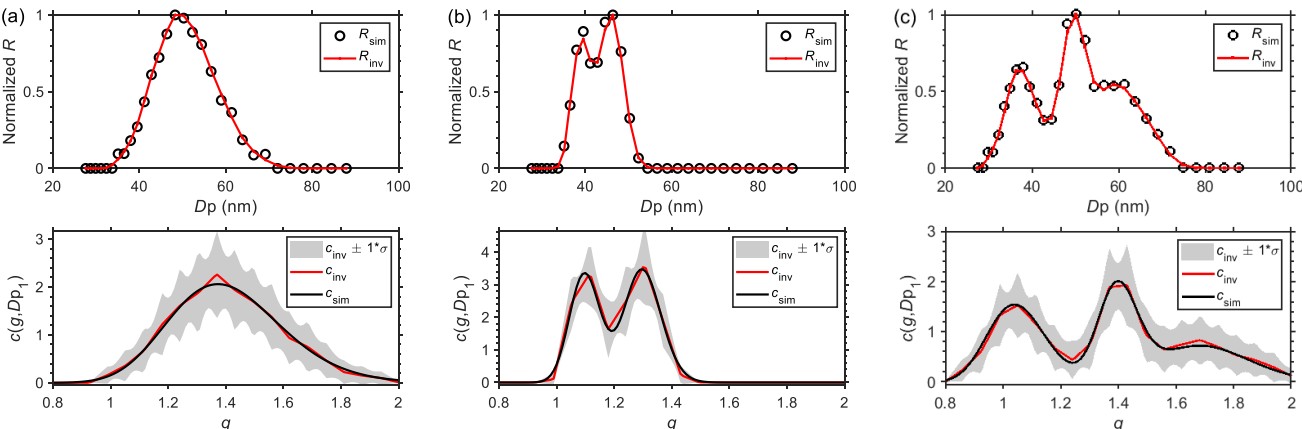

**Figure 6.** (*Top panels*) Comparisons between the averaged reconstructed HFIMS measurements and the simulated HFIMS measurements corrupted with Poisson noises for pre-defined GF-PDFs of one mode (**a**), two modes (**b**), and three modes (**c**), respectively. (*Bottom panels*) Comparisons between the pre-defined GF-PDFs and the GF-PDFs inverted using Twomey's method with the optimized value of GF bins.

The shaded area represents GF-PDF solution spaces within one standard deviation.

To quantify the accuracy of the inverted GF-PDFs, we fitted the inverted GF-PDFs to recover the mode parameters as shown in Table 2. The pre-set mode parameters of the pre-defined GF-PDFs are shown in Table 1. The results show that both the mode geometric means and the multimodal number fractions can be recovered accurately with minor uncertainties.

**Table 2.** Recovered mode parameters of pre-defined GF-PDFs from inverted GF-PDFs.

| Predefined | Mode 1 | | | Mode 2 | | | Mode 3 | | |
|---|---|---|---|---|---|---|---|---|---|
| GF-PDF | $f$ | $G$ | $\sigma$ | $f$ | $G$ | $\sigma$ | $f$ | $G$ | $\sigma$ |
| 1 | $1.00 \pm 0$ | $1.39 \pm 0.03$ | $1.09 \pm 0.01$ | | NA | | | NA | |
| 2 | $0.46 \pm 0.10$ | $1.10 \pm 0.02$ | $1.02 \pm 0.01$ | $0.54 \pm 0.10$ | $1.30 \pm 0.02$ | $1.03 \pm 0.01$ | | NA | |
| 3 | $0.37 \pm 0.09$ | $1.05 \pm 0.03$ | $1.05 \pm 0.02$ | $0.34 \pm 0.13$ | $1.40 \pm 0.03$ | $1.03 \pm 0.02$ | $0.28 \pm 0.12$ | $1.69 \pm 0.08$ | $1.06 \pm 0.03$ |

## 4 Conclusion

In this study, we develop and evaluate nonparametric regularized methods for inverting GF-PDF from HFIMS measurements. The integrated response of HFIMS, which is a convolution of the aerosol hygroscopic GF-PDF, the transfer function of the

400 DMA, and the transfer function of the WFIMS, is first cast into a matrix form. With the matrix form, nonparametric regularized methods can be applied straightforwardly to invert the GF-PDF. Synthetic HFIMS measurements are generated using Monte-Carlo simulations for representative aerosol GF-PDFs, and the synthetic data are used to investigate the dependence of inverted

GF-PDF on the number of GF bins (i.e., GF resolutions) and the performances of different inversion methods. We show an optimal GF bin number of 20 for all nonparametric methods and representative GF-PDFs. The performances of unregularized least-squares, Twomey's algorithm, Tikhonov regularizations, and commonly used parametric inversion methods (i.e., ML and PL least-squares fitting) are compared. Nonparametric methods based on the matrix form have substantial advantages in the inversion of GF-PDF over the parametric fitting methods as (1) no prior assumption of GF-PDF distributions is required; (2) the matrix-based form facilitates the application of different regularizations (e.g., Tikhonov regularization and Twomey's iterative regularization), which reduce the error in inverted GF-PDF by eliminating noise amplification; (3) they are much more computationally efficient once the matrix is generated. The Tikhonov regularized solutions depend on the regularization parameter, $\lambda$. While the Tikhonov regularization can outperform Twomey's method in theory, the optimal $\lambda$ value cannot be obtained reliably using existing methods in practice, leading to inferior performances than Twomey's method. On average, the GF-PDF inverted using Twomey's method has the smallest error compared to solutions using the other inversion methods regardless of the shapes of the pre-defined GF-PDFs, and it accurately reproduces the true GF-PDF, including the mode parameters and other key statistics.

*Data availability*. Datasets and code packages related to this paper is provided in a GitHub repository (https://github.com/zjs023/Regularized_inversion_HFIMS). More information is provided with the README in the repository.

*Author contributions*. JZ and JW designed the study. JZ developed the code. JZ and JW prepared the manuscript with contributions from all co-authors.

*Competing interests*. The authors declare that they have no conflict of interest.

*Acknowledgements.* We acknowledge the funding support from the U. S. Department of Energy's Small Business Innovation Research (SBIR) program under contract DE-SC0013103 and Small Business Technology Transfer (STTR) program under contract DESC0006312.

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
