# Peer review of "Regularized inversion of aerosol hygroscopic growth factor probability density function: Application to humidity-controlled fast integrated mobility spectrometer measurements"

_Atmospheric Measurement Techniques, 2021_

## Referee Comment (RC1)

**Review to "Regularized inversion of aerosol hygroscopic growth factor probability density function: Application to humidity-controlled fast integrated mobility spectrometer measurements", Atmos. Meas. Tech., December 2021**

The authors test of different inversion methods for aerosol hygroscopicity measurements with a specific focus on the humidity-controlled fast integrated mobility spectrometer. The manuscript is very well written, and concise. As inversion techniques other than least-square approaches have recently gained some attention in the community again (Petters, 2021; Sipkens et al., 2020a, 2020b; Stolzenburg et al., 2022), this manuscript certainly is another interesting approach focusing on an important problem, the retrieval of the GF-PDF. However, with respect to the above-mentioned literature, I have the feeling that the authors need to add additional work to their manuscript, such that it significantly adds to the body of knowledge and I can recommend publication in Atmos. Meas. Tech. I give my detailed suggestions below:

1) The authors need to challenge their inversion technique with more difficult data. Using the same forward and inverse model is considered to be not sufficient in testing an inversion approach, leading to unrealistically good outputs (e.g. Colton and Kress, 2013). The authors need to run tests on their inversion where the forward and inverse model are different (i.e. assuming that the calibration of the FIMS is not perfect) and where statistics other than the counting error influence the result and their actual distribution can only be guessed (e.g. the standard deviation of an additional Gaussian error is different in forward and inverse model). Both cases are closer to real measurements and this would demonstrate the performance of the inversion algorithms under more challenging conditions.
2) The authors should at least present an application of their algorithms to real measurement data, such that the reader can judge how big differences we would expect in the retrieval of the GF-PDF for typical hygroscopicity measurements in the ambient air.
3) *Data availability*. I think it is nowadays almost standard that the authors publish their code openly along with the corresponding manuscript (again referring to Petters, 2021; Sipkens et al., 2020b; Stolzenburg et al., 2022). If the authors want to ensure reuse of their methods this is highly encouraged.
4) I understand that the manuscript focuses on the inversion for the FIMS, but this instrument is not available to many researchers doing hygroscopicity studies. In my opinion, it would be highly beneficial if the authors could also test their inversion on classical TDMA data.

Apart from the above, I have only a couple of minor comments:

1) L.153 ff.: Please note that Tikhonov regularization (and LSQ) implicitly assume Gaussian statistics on the measurement noise, but that the underlying statistics used to generate the measurement data is of Poisson nature in this manuscript (see also comment above on imposing an additional Gaussian uncertainty to challenge the methods under more difficult conditions). You can refer to Stolzenburg et al. (2022) here who developed a Poisson approach for regularization.
2) L.159: Please be aware that the matrices described here impose specific boundary conditions (Donatelli and Reichel, 2014). I assume that Dirichlet boundary conditions have been used. Please specify this, as it is often neglected in inversion procedures that the boundary condition is quite important for the shape of the solution.
3) L.161 ff.: Which algorithm was used to locate the L-curve corner?
4) L.242: The discussion about the computing time is difficult, as it crucially relies on the used device. I think it is better to make Figure 2d with relative computing times.

5) L.250 ff.: The measure smoothness is defined in Eq. (14) by the $2^{nd}$ order derivative, so it is not surprising to me, that the $2^{nd}$ order Tikhonov regularization achieves the best smoothness measure, as it optimizes for the same quantity.

6) L. 253: It is somewhat expected that the absolute values of $\xi$ increase with more modes. The more interesting question is rather why it is not so significant for the Twomey algorithm.

References:

Colton, D. and Kress, R.: Inverse Acoustic and Electromagnetic Scattering Theory, 3rd ed., Springer-Verlag, New York., 2013.

Donatelli, M. and Reichel, L.: Square smoothing regularization matrices with accurate boundary conditions, J .Comput. Appl. Math., 272, 334–349, doi:https://doi.org/10.1016/j.cam.2013.08.015, 2014.

Petters, M. D.: A Software Package to Simplify Tikhonov Regularization with Examples for Matrix-Based Inversion of SMPS and HTDMA Data, Atmos. Meas. Techn. Discuss., 2021, 1–27, doi:10.5194/amt-2021-51, 2021.

Sipkens, T. A., Olfert, J. S. and Rogak, S. N.: Inversion methods to determine two-dimensional aerosol mass-mobility distributions: A critical comparison of established methods, J. Aerosol Sci., 140, 105484, doi:https://doi.org/10.1016/j.jaerosci.2019.105484, 2020a.

Sipkens, T. A., Olfert, J. S. and Rogak, S. N.: Inversion methods to determine two-dimensional aerosol mass-mobility distributions II: Existing and novel Bayesian methods, J. Aerosol Sci., 146, 105565, doi:https://doi.org/10.1016/j.jaerosci.2020.105565, 2020b.

Stolzenburg, D., Ozon, M., Kulmala, M., Lehtinen, K. E. J., Lehtipalo, K. and Kangasluoma, J.: Combining instrument inversions for sub-10 nm aerosol number size-distribution measurements, J. Aerosol Sci., 159, 105862, doi:https://doi.org/10.1016/j.jaerosci.2021.105862, 2022.

---

## Author Comment (AC2)

Manuscript No.: amt-2021-334

**Title**: Regularized inversion of aerosol hygroscopic growth factor probability density function: Application to humiditycontrolled fast integrated mobility spectrometer measurements

5 We thank the anonymous referees for their valuable and constructive comments/suggestions on our manuscript. We have revised the manuscript accordingly and please find our point-to-point responses below.

**Comments by Anonymous Referee #2:**

**General Comments:**

- 10 The authors apply different inversion methods to invert the aerosol GR-PDF from the measured signals from synthetic HFIMS signals. They found that for the few test cases, Markowski-Towmey's method generally outperforms other methods. By doing this, they convincingly improved the data inversion of HFIMS data and promisingly HTDMA data, which were mainly based on predefined size distributions or least square methods. This well-written manuscript is easy to follow. I recommend it to be published in Atmospheric Measurement Techniques, However, a major revision is necessary to convincingly demonstrate that
- 15 the data inversion of HFIMS (and HTDMA) is improved. I feel that the authors are too optimistic about the representativity of their limited synthetic data on real laboratory experiments and atmospheric measurements. Further, this manuscript will have a broader impact on the community if its outcomes (e.g., inversion codes) can be readily used for HTDMA measurements. My detailed comments are given below.

**Responses**: We thank the reviewer for the constructive suggestions and comments. Point-to-point responses to comments and

20 questions are detailed below. Following the reviewer's suggestions, we examined the impact of additional Gaussian noises on the performance of inversion methods. We also applied the inversion algorithms to ambient measurement data and compared the performances. Additional analyses were also carried out to elucidate why Twomey's method is statistically better than the Tikhonov regularization methods. The new results and discussions are now included in the revised manuscript.

**25 Detailed Comments:**

1) More tests and/or discussions are needed to provide supports for the argument that Towmey's method outperforms other tested inversion methods. The three test cases are perhaps sufficient to show that Towmey's method is better than least square methods because the least square methods are notorious for solving ill-conditioned problems. However, the reason why Towmey's method is better than the Tikhonov regularization methods needs more clarification and/or data to support.

30 **Responses**: We thank the reviewer for this comment. To elucidate why Twomey's method is better than the Tikhonov regularization methods, we compare the results of inversions based on Twomey's method and 1st order Tikhonov regularization with the regularization parameter derived using three different approaches. The first approach (i.e., the L-curve approach, Hansen and O'Leary, 1993), derives the  $\lambda$  by seeking a trade-off between minimizing the residual term and minimizing the regularization term (i.e., roughness of the solution). The 2nd approach is based on the Hanke-Raus rule, which selects the  $\lambda$

- 35 value that minimizes the λ-dependent residual term  $\frac{1}{\lambda} \|\mathbf{M}\mathbf{c}^{Tik}(\lambda) \mathbf{R}\|_2$  (Hanke and Raus, 1996; Sipkens et al., 2020). In the 3rd approach, the value of λ is optimized by comparing the inverted GF-PDF with the correct solution, i.e., minimizing the Euclidean distance between the inverted and the pre-defined GF-PDF. Therefore, inversion based on λ derived using the 3rd approach represents the best possible performance of the Tikhonov regularization. It is worth noting that the 3rd approach is not possible for real measurements, as the true GF-PDF is unknown. As shown in Fig. 1, the Tikhonov regularized solution
- 40 strongly depends on the regularization parameter  $\lambda$ . The Tikhonov regularization with the optimized  $\lambda$  (derived using approach #3) provides the most accurate solution (i.e., lowest GF-PDF error) as expected, and outperforms Twomey's method. However, when  $\lambda$  derived using the L-curve approach or the Hanke-Raus rule is used, GF-PDF inverted using 1st order Tikhonov regularization generally has a larger error (i.e.,  $\gamma^2$ ) than that inverted using Twomey's method. The above comparisons indicate that while Tikhonov regularization can outperform Twomey's method in theory, the optimal regularization parameter  $\lambda$  cannot
- 45 be obtained reliably using existing methods in practice, leading to inferior performance than Twomey's method. For example, the L-curve approach does not work well if the curvature of the L-curve is negative everywhere, and in such scenario, the leftmost point (i.e., with smaller  $\lambda$ ) on the L-curve is taken as the corner (Hansen, 1994), leading to insufficient regularizations of the solution (Naseri et al., 2021). On the other hand, the Hanke-Raus rule often chooses a much larger  $\lambda$  compared with the optimal value, which results in over-smoothed solutions potentially with even larger errors. We also carried out similar

50 comparisons of Twomey's method with 0th and 2nd order Tikhonov regularizations with  $\lambda$  values derived using the three different approaches, and the results are consistent. We have included the above comparison and discussion in the revised manuscript (line 335-370).

Figure 1. The reconstruction residual,  $\chi^2$  (a), the GF-PDF error,  $\gamma^2$  (b), and the smoothness,  $\xi$  (c) of GF-PDF inverted using LSQ, 1st order Tikhonov regularization with the regularization parameter derived from three different approaches (L-curve, Hanke-Raus rule, and optimized  $\lambda$ ), and Twomey's method. The colors correspond to the pre-defined GF-PDFs with one mode (blue), two modes (orange), and three modes (yellow). The results are averages based on inversions of 500 sets of synthetic HFIMS data for each of three pre-defined GF-PDFs.

2) The authors need to show the performance of Towmey's method with at least one dataset from either laboratory experiments
or atmospheric measurements. Estimating the measurement uncertainties with only the counting uncertainties typically

underestimates the total uncertainties. Despite this, I am not concerned about the applicability of Towmey's to real datasets and its better performance of the than least square methods.

- Responses: Following the reviewer's suggestion, we apply the nonparametric inversion methods to ambient HFIMS measurements, and the results are compared in Fig. 2. The HFIMS responses reconstructed from GF-PDF inverted using unregularized LSQ, Tikhonov, and Twomey's methods generally match the measurement (black circle) well. The GF-PDF at 85% RH for ambient 35 nm particles consist of a smaller less-hygroscopic mode and a larger more-hygroscopic mode. As expected, the HFIMS response reconstructed from LSQ inverted GF-PDF has the minimum deviation from the actual measurement whereas the GF-PDF exhibits more oscillations near the tail of the second mode. These oscillations create a small third mode that is absent from the smoother GF-PDFs inverted using regularized methods (i.e., Tikhonov and Twomey's methods). GF-PDF inverted using Twomey's method and 0th Tikhonov clearly distinguish the two growth factor modes. In comparison, the two modes become more overlapped in GF-PDF inverted using 1st and 2nd Tikhonov regularization, due to additional and possibly excessive regularization.
  - Rmeas LSQ (a) ο (b) 6 Tik 0th Rinv, LSQ 0.8 Tik\_1st R inv, Tik\_0th Tik 2nd 5 Normalized R Twomey R inv, Tik\_1st 0.6  $c(g, Dp_1)$ R inv, Tik\_2nd R inv, Twomey 0.4 3 2 0.2 0 0 30 40 50 60 70 80 0.8 1.2 1.4 1.6 1.8 2 1  $D_{\rm p}$  (nm) g

**Figure 2.** (a) Comparison between the HFIMS measured response (black circle) and the responses (marked lines) reconstructed from GF-PDF derived using different methods for 35 nm ambient aerosol at 85% RH. (b) Inverted GF-PDFs using different methods.

We also examined the statistics of the reconstruction residual and the smoothness of GF-PDF inverted from 3-day HFIMS measurements using the listed nonparametric methods. Among all nonparametric inversion methods, unregularized LSQ leads to the lowest reconstruction residual but the worst smoothness (Fig. 3). As regularizations are introduced in the Tikhonov algorithms, the inverted GF-PDFs become smoother at the expense of increased reconstruction residuals. The Tikhonov regularized solutions strongly depend on the regularization parameter  $\lambda$ . In this study, the value of  $\lambda$  has been derived using three approaches, including (1) the L-curve, (2) the Hanke-Raus rule, and (3) comparison of inverted GF-PDF with the true solution. Note the 3rd approach (i.e., comparison of inverted GF-PDF with the true solution) is not possible for ambient

85 measurements. Inversions of synthetic data show that the L-curve approach generally underestimates the regularization

parameter (Fig. 5 in the manuscript), resulting in insufficiently regularized solutions (Naseri et al., 2021). For the 3-day ambient measurements, when  $\lambda$  is derived using the L-curve approach, the reconstruction residuals for the GF-PDF inverted using Tikhonov algorithms are very close to those of the unregularized LSQ, consistent with underestimated  $\lambda$  values (Fig. 3a and d). In contrast, Tikhonov regularizations with  $\lambda$  value determined using the Hanke-Raus rule tend to over-smooth solutions

- 90 due to overestimated  $\lambda$  values, resulting in significantly increased errors in reconstructed HFIMS measurements (Fig. 3b and e). The 3-day ambient measurements are also inverted using Tikhonov algorithms with an empirical  $\lambda$  value of 0.03 (Fig. 3c and f), which corresponds to the mean value of optimized  $\lambda$  values (i.e., derived using the 3rd approach) for the synthetic HFIMS data. The inverted GF-PDF shows improved smoothness compared to the solution from the LSQ method, without introducing excessive reconstruction errors. While the empirical  $\lambda$  value appears to work quite well for the 3-day
- 95 measurements, using this fixed regularization parameter may not be appropriate for other ambient measurements. For Twomey's method, both the reconstruction residual and the smoothness are between those based on the 0th order and 1st order Tikhonov regularizations with the empirical regularization parameter ( $\lambda = 0.03$ ), suggesting an appropriate trade-off between the GF-PDF smoothness and the fidelity in reproducing the HFIMS measurements. Note that the statistics of the GF-PDF error cannot be derived as the actual GF-PDF of ambient aerosols are unknown. As a result, it is difficult to draw a definite
- 100 conclusion regarding which method has the best performance in retrieving the GF-PDF based on the ambient measurements. The above results and discussion have been added in Section S5 of the revised supplemental information.

---

## Author Comment (AC3)

Manuscript No.: amt-2021-334

**Title**: Regularized inversion of aerosol hygroscopic growth factor probability density function: Application to humiditycontrolled fast integrated mobility spectrometer measurements

5 We thank the anonymous referees for their valuable and constructive comments/suggestions on our manuscript. We have revised the manuscript accordingly and please find our point-to-point responses below.

**Comments by Anonymous Referee #1:**

**General Comments:**

- 10 The authors test of different inversion methods for aerosol hygroscopicity measurements with a specific focus on the humidity-controlled fast integrated mobility spectrometer. The manuscript is very well written, and concise. As inversion techniques other than least-square approaches have recently gained some attention in the community again (Petters, 2021; Sipkens et al., 2020a, 2020b; Stolzenburg et al., 2022), this manuscript certainly is another interesting approach focusing on an important problem, the retrieval of the GF-PDF. However, with respect to the above-mentioned literature, I have the
- 15 feeling that the authors need to add additional work to their manuscript, such that it significantly adds to the body of knowledge and I can recommend publication in Atmos. Meas. Tech. I give my detailed suggestions below.

**Responses**: We thank the reviewer for the constructive comments and suggestions. Point-to-point responses to comments and questions are detailed below. Following the reviewer's suggestions, we examined the impact of additional Gaussian

20 noises on the performance of inversion methods. The inversion methods were also tested with different forward and inverse models (i.e., assuming that the calibration of the FIMS/DMA is not perfect). We also applied the inversion algorithms to ambient measurement data and compared the performances. The new results and discussions are now included in the revised manuscript.

**25 Detailed Comments:**

1) The authors need to challenge their inversion technique with more difficult data. Using the same forward and inverse model is considered to be not sufficient in testing an inversion approach, leading to unrealistically good outputs (e.g. Colton and Kress, 2013). The authors need to run tests on their inversion where the forward and inverse model are different (i.e. assuming that the calibration of the FIMS is not perfect) and where statistics other than the counting error influence the

30 result and their actual distribution can only be guessed (e.g. the standard deviation of an additional Gaussian error is different in forward and inverse model). Both cases are closer to real measurements and this would demonstrate the performance of the inversion algorithms under more challenging conditions.

**Responses**: The forward model and inverse model are not exactly identical. First, in the forward calculation, the HFIMS measurement  $R_i$  (i.e., the convolution of the GF-PDF, the DMA transfer function, and the WFIMS transfer function) is

35 integrated using Eq. (4) with a much higher resolution of g (i.e., 120 bins over 0.8 - 2.0) while the inverse model casts the HFIMS kernel into a matrix with only 20 bins of g. In addition, the synthetic HFIMS measurements include noise due to counting statistics based on Poisson distribution.

Following the reviewer's suggestion, we also challenged our algorithm with different forward and inverse models to simulate the scenarios when DMA or WFIMS is not perfectly calibrated. The particle sizes measured by DMA and WFIMS

- 40 are determined by the voltage and sheath flow, which can be calibrated straightforwardly. Therefore, the nonideality in DMA and WFIMS performance likely manifests in the deviation of instrument mobility resolution from the theoretical value. To test the performance of inversion algorithms for such scenarios when the transfer function of DMA or WFIMS is not fully calibrated, we generate the synthetic HFIMS measurements by perturbing DMA or WFIMS mobility resolution (i.e.,  $R_{DMA}$  or  $R_{WFIMS}$ ), while maintaining the theoretical  $R_{DMA}$  or  $R_{WFIMS}$  in the inverse model. The mobility resolution is perturbed by
- 45 varying the ratio of sheath to aerosol flow for DMA or WFIMS ( $R_Q=Q_{sh}/Q_a$ ) in the derivation of the transfer function. The default flow rate ratio for DMA and WFIMS are 10 and 50, respectively. Figures 1 and 2 show the inversion results when DMA  $R_Q$  in the forward model is varied from 8 to 12 while WFIMS  $R_Q$  is maintained at the actual value of 50 and when DMA  $R_Q$  is maintained at 10 while WFIMS  $R_Q$  is varied from 40 to 60. The results are based on inversions of 500 sets of synthetic HFIMS measurements (with the noise of counting statistics included) using Twomey's method. The average
- 50 residual ( $\chi^2$ ), the GF-PDF error ( $\gamma^2$ ), and the smoothness ( $\xi$ ), all showed very minor variation with DMA or WFIMS  $R_Q$ used in the forward model, suggesting negligible impacts on Twomey inversion results due to imperfect calibration of DMA and WFIMS resolution. A possible explanation is that typical GF-PDFs of ambient aerosol particles are relatively broad such that the inverted GF-PDF is insensitive to DMA and WFIMS resolution. The impact of  $R_Q$  on other nonparametric methods was also investigated and found negligible.

55

Figure 1. The reconstruction residual,  $\chi^2$  (a), the GF-PDF error,  $\gamma^2$  (b), and the smoothness,  $\xi$  (c) of GF-PDF inverted using Twomey's method as a function of DMA  $R_Q$  used to calculate DMA transfer function in the forward model (WFIMS  $R_Q$  maintained at the actual value of 50). Actual DMA and WFIMS  $R_Q$  values of 10 and 50 are used to derive transfer functions in the inverse model (i.e., calculation of the inversion matrix). The colors correspond to the pre-defined GF-PDFs with one mode (blue), two modes (orange), and three modes (yellow). The results are averages based on the inversion of 500 sets of synthetic HFIMS data for each of three pre-defined GF-PDFs.

60

**Figure 2.** The reconstruction residual,  $\chi^2$  (**a**), the GF-PDF error,  $\gamma^2$  (**b**), and the smoothness,  $\xi$  (**c**) of GF-PDF inverted using Twomey's method as a function of WFIMS  $R_0$  used to calculate WFIMS transfer function in the forward model (DMA  $R_0$  maintained at the actual value of 50). Actual DMA and WFIMS  $R_Q$  values of 10 and 50 are used to derive transfer functions in the inverse model (i.e., calculation of the inversion matrix). The colors correspond to the pre-defined GF-PDFs with one mode (blue), two modes (orange), and three modes

65

(yellow). The results are averages based on the inversion of 500 sets of synthetic HFIMS data for each of three pre-defined GF-PDFs.

In addition to noise due to counting statistics, we also included additional Gaussian noise (e.g., due to variations of sample flow rate) ranging from 1% to 10% in generating the synthetic HFIMS data and examined the impact of the additional noise on inverted GF-PDFs. Figure 3 shows that Gaussian noises up to 10% have negligible impact on the reconstruction residual  $(\gamma^2)$ , the error  $(\gamma^2)$ , and the smoothness  $(\xi)$  of GF-PDFs inverted using Twomey's method. Similarly, the impact is also negligible for GF-PDF inverted using unweighted LSQ and 0th, 1st, and 2nd order Tikhonov regularizations (not shown). The negligible impacts indicate that the noise of typical HFIMS measurements is dominated by counting statistics. The above results and discussion are detailed in a new section (Section 3.2) titled "Effect of measurement uncertainties" in the revised 75 manuscript (line 253-299).